# Proposal of Model for Evaluation of Viral Kinetics of African/Asian/Brazilian—*Zika virus* Strains (Step Growth Curve) in Trophoblastic Cell Lines

**DOI:** 10.3390/v15071446

**Published:** 2023-06-27

**Authors:** Márcia Duarte Barbosa, Anderson Costa, Paula Prieto-Oliveira, Robert Andreata-Santos, Cristina M. Peter, Paolo M. A. Zanotto, Luiz Mario Ramos Janini

**Affiliations:** 1Laboratory of Molecular Evolution and Bioinformatics, Department of Microbiology, Institute of Biosciences, University of São Paulo, São Paulo 05508-000, Brazil; pzanotto@usp.br; 2Laboratory of Retrovirology, Department of Microbiology, Immunology and Parasitology, Discipline of Microbiology, Federal University of São Paulo, São Paulo 04039-032, Brazil; andersonstaphy@gmail.com (A.C.); randreatas@gmail.com (R.A.-S.); cristinamendespeter@gmail.com (C.M.P.); janini@unifesp.br (L.M.R.J.); 3Department of Bioinformatics and Genomics, College of Computing and Informatics, University of North Carolina at Charlotte, 9331 Robert D. Snyder Rd., Charlotte, NC 28223, USA; polivei1@charlotte.edu

**Keywords:** *Zika virus*, trophoblast cell, viral kinetics

## Abstract

The *Zika virus* (ZIKV) epidemic brought new discoveries regarding arboviruses, especially flaviviruses, as ZIKV was described as sexually and vertically transmitted. The latter shows severe consequences for the embryo/fetus, such as congenital microcephaly and deficiency of the neural system, currently known as Congenital ZIKV Syndrome (CZS). To better understand ZIKV dynamics in trophoblastic cells present in the first trimester of pregnancy (BeWo, HTR-8, and control cell HuH-7), an experiment of viral kinetics was performed for African MR766 low passage and Asian-Brazilian IEC ZIKV lineages. The results were described independently and demonstrated that the three placental cells lines are permissive and susceptible to ZIKV. We noticed cytopathic effects that are typical in in vitro viral infection in BeWo and HTR-8. Regarding kinetics, MR766lp showed peaks of viral loads in 24 and 48 hpi for all cell types tested, as well as marked cells death after peak production. On the other hand, the HTR-8 lineage inoculated with ZIKV-IEC exhibited increased viral production in 144 hpi, with a peak between 24 and 96 hpi. Furthermore, IEC had peak variations of viral production for BeWo in 144 hpi. Considering such in vitro results, the hypothesis that maternal fetal transmission is probably a way of virus transmission between the mother and the embryo/fetus is maintained.

## 1. Introduction

According to the (ICTV, 2023) classification, ZIKV belongs to the *Flavivirus* genus and the *Flaviviridae* Family. Its genome is presented as a positive-sense single-strand RNA (ssRNA+) that has around 11 kb. The genomic RNA is constituted by one strand and presents an ORF that codes for a single polyprotein. When it is cleaved, it produces structural and non-structural proteins.

*Zika virus* was discovered in 1947 during a research expedition on Yellow Fever in the Ziika Forest located in Uganda, Africa [1,2,3]. Only in 1964 did, Simpson et al. report the first human case of ZIKV infection in Uganda [4]. In 1966, the presence of the virus in the *Aedes aegypti* mosquito circulating in the urban area was registered, for the first time, in Malaysia [5]. Furthermore, the first recorded outbreak of Zika Fever occurred between 1977 and 1978 in Indonesia [6,7]. Until then, the infection was only identified by a serological test in humans [7]. Between 1947 and 2007, Posen et al., (2016) reported the presence of the ZIKV in twenty-nine African, eight Asian, and one European countries [8,9,10]. Between 2013 and 2016, ZIKV spread to a larger global circulation, being the causal agent for several outbreaks of Zika Fever in 2016 [8,9,10]. Brazil was one of the most affected countries and registered one of the highest numbers of infected people, with Guillain Barre Syndrome (GBS) in adults and congenital ZIKV Syndrome (CZS) in newborns [9,10,11,12].

The major ZIKV transmission is vectorial, with a focus of infection on the nervous system [9,13]. Considering the viral detection in the amniotic fluid of pregnant women and in the female and male sexual organs, it is possible that ZIKV can be transmitted by means of sexual relations and maternal fetal attachment [13,14,15,16,17,18]. Among flaviviruses, there are cases of vertical transmission (maternal fetal) for CHIKV, DENV, and WNV [8,9], and WNV can infect trophoblast cells more efficiently when compared with ZIKV [19]. However, its relevance for the spread and disease of such viruses is still not conclusive. ZIKV cell tropism includes the central and peripheral nervous system, epithelial cells, immune system, and reproductive system [8,9,13]. In placental tissues, ZIKV has been identified in syncytiotrophoblasts, cytotrophoblasts, and endothelial cells of the maternal immune system (dendritic cells and macrophages) [14]. A receptor for ZIKV is present in female reproductive tissues and in the placenta; the receptors are the Gas6-AXL receptor tyrosine kinase complex and AXL tyrosine kinase protein [15]. Another candidate receptor for virus entry into trophoblastic cells is FcRn [16].

The Step Growth Curve (SGC) methodology was defined as standard to observe virus behavior over time, since such can be used in silico, in vitro, in/ex situ, and in/ex vivo [20,21,22,23,24]. Moreover, the data obtained through SGC allows application in different studies, such as in vaccines, antivirals, epidemiology, viral evolution, comparison between strains/variants/serological of the same species or different species, double or more infections, and intra- and inter- viral species in the same host, among others. Since sexual and maternal-fetal transmission is uncommon for Flaviviruses, this study aimed to evaluate selected BeWo and HTR-8 placental cells to verify whether the Asian-Brazilian strain of ZIKV has the same behavioral dynamics as the original African strain in sustained replication (viral kinetics).

## 2. Materials and Methods

### 2.1. Viral Isolation and Formation of Viral Stocks

ZIKV-MR766lp. Low-passage (lp) strain isolated from *Macaca mulatta* in 1947 in Ziika Forest, Uganda, Africa (GenBank: AY632535.2) [25,26,27,28,29]. It presents the term low passage because it has a smaller number of passages in mice, according to information provided by Dr. Amadou Alpha Sall, from the Pasteur Institute of Dakar, Senegal, Africa, who kindly provided samples for this study.

ZIKV-IEC-Paraíba. Strain isolated from a patient in Paraíba, Brazil, clinically diagnosed with ZIKV in the 2015 outbreak (GenBank: KX2800260) [25,26,27,28,29]. The sample was kindly provided by Prof. Dr. Pedro Vasconcelos from the Evandro Chagas Institute in Belém, Pará. 

Both strains were cultivated in Vero CCL-81 cells (ATCC^®^) to form viral stocks and to get obtain the necessary amount for complete experimentation. Such stocks were produced in three steps to obtain the maximum yield of infective virions. The first step involved cultivation in five cylindrical tubes of glass with a volume of 1 mL of cells. The second cultivation step occurred in five culture flasks with a 25 cm^2^ Corning^®^ filter and volumes between 5 and 7 mL, and the third occurred in five culture flasks with a 75 cm^2^ Corning^®^ filter and volumes between 15 and 20 mL. For each step, the supernatant, debris, and cell monolayer were collected together; gathered in a single container; and homogenized to be stored in a 1.8 mL Corning^®^ cryotube, frozen in a freezer at −80 °C. We separated a small aliquot from the three steps for titration by PFU/mL and tested for the presence of mycoplasma according to Timenetsky et al. [30].

### 2.2. Cells Cultures

For Vero CCL-81, HTR-8/SVneo, and BeWo, we followed the cell maintenance as described by the ATCC^®^ cell bank, while for HuH-7, JCRB Cell Bank recommendations were used. Cellular, viral stocks, and samples from viral kinetics experiments were tested for mycoplasma [30] and RNA contaminations by ZIKV [31]. All results were negative.

BeWo is a fusiogenic choriocarcinoma cell type—it forms syncytium with human villous trophoblastic cells properties. It shows features common to normal trophoblasts, in addition to expressing IL-6, IL-10, IFN-α, IFN-β, hCG, steroids, estrogens, and progesterone. For this reason, it is a good model to study the dynamics of viral infection (ATCC^®^). Considering that BeWo in vitro has a low rate of spontaneous fusion, we used forskolin (chemical compound that accelerates the fusiogenic capacity of such a cell lineage) [32].

HTR8/SVneo is a cell type of human extravillous trophoblast immortalized by SV40 virus, and originates from chorionic villi, present between 6 and 12 weeks of gestation (first gestational trimester). Furthermore, it is characterized as a type of epithelial cell, with hCG production and invasion of maternal uterine tissue (ATCC^®^).

Both cell lineages were kindly provided by Profa. Dr. Estela M. A. F. Bevilacqua, from the Maternal Fetal and Placenta Interaction Studies Laboratory, ICB-USP, São Paulo, SP, Brazil (Appendix B).

Vero CCL-81 is a type of epithelial adherent cell from the kidney of a normal adult monkey that belongs to the African species *Chlorocebus aethiops* (ATCC^®^).

HuH-7 (JCRB0403) is an adherent cell lineage that is immortalized and originates from the tumorigenic epithelium of *Homo sapiens*—a hepatocellular carcinoma commonly used for HCV studies (JCRB).

### 2.3. Standard Curve for PFU/mL Determination

The correlation between C_T_ and PFU/mL or a standard curve was established through titration by PFU/mL performed in triplicate for ZIKV-MR766 and ZIKV-IEC, in plates of in 24-well with a cell concentration of 1 × 10^5^ cells/mL per well of Vero cells. Moreover, the stock was titrated using 0.2 mL of the viral sample in a serial dilution of 10-fold (10^−1^ to 10^−11^). Then, we incubated the plates for five days in an oven with a temperature at 37 °C and 5% of CO_2_. Thereafter, the supernatant was collected and stored for quantification by qRT-PCR (Appendix C) and the cells monolayers were fixed to determine the viral titer. Based on a conversion table, C_T_ values were calculated and obtained from viral titers in PFU/mL (Appendix I).

### 2.4. Viral Kinetics (Step Growth Curve)

In this experiment, we verified the capacity of infectivity and viral multiplication that both viral strains of ZIKV (African and Asian-Brazilian) have, considering the following cell lineages: BeWo, BeWo treated with forskolin, HTR-8, and HuH-7.

The step growth curve presented seven time points, determined as hour post infection (hpi), starting from two hpi in the incubator and a 24 h interval until 144 h. Therefore, we obtained the following hpi: 2 h, 24 h, 48 h, 72 h, 96 h, 120 h, and 144 h. Sample collection for each time point was performed in duplicate, with the separation of samples between the supernatant and cell monolayer, which was stored in a freezer at −80 °C for later analysis (Appendix D). The multiplicity of infection (MOI). It is the ratio of the number of viral particles to the number of host cells. The MOI = 1 implies that for each cell unit, there is a single viral particle. The encounter of a particle by the host cell is a chance encounter, therefore statistically it can be represented by a Poisson distribution [20,21,22,23,24]. The variation was determined between 0.1 and 1 due to the different replicative capacity of both strains. According to the Poisson distribution, there are few differences within this variation when applied in the assay.

The data obtained from the first complete growth curve were used as standard for the analysis of the two remaining replicates at 24, 48, and 72 hpi. Thereafter, we calculated PFU/mL for each time point from qRT-PCR data based on the standard curve, constructed before the beginning of the kinetics experiment (Appendix E and Appendix I). Then, a description of the cytopathic effect was noticed for all time points of three replicates. Since it was not possible to fulfill the replications at the same time and in the same passage of cells, we decided that the cells had two passages of difference for each replication of the kinetic assay. Images for all kinetics were seen on the Thermo Fisher Scientific EVOS™ FL (Waltham, MA, USA) inverted light microscope at 20× (200 μM). The viral growth curve is based on the theoretical curves of Delbruck and Ellis [20] and Burleson [23] (Figure 1, step 3).

## 3. Results

Uninfected cells (BeWo, HTR-8, and HuH-7) were observed and compared with infected cells at each time point. Cell viability was considered visually, with the support from data published in the literature [14,16,19,33,34,35] and from cell banks. Images of mock and infected cells can be seen in the Appendix A Appendix A.

For the standard curve, we used the average of the titers to construct the standard curve for each strain—conversion from C_T_ to PFU/mL (Appendix E, Appendix F, Appendix G and Appendix A).

Viral Kinetics ZIKV-MR766lp. The titer of the viral stocks was 1.7 × 10^8^ PFU/mL. We utilized MOI = 1 and it were found the following averages of cell concentration were found: BeWo = 3.14 × 10^6^ cell/mL, BeWo + fork = 1.7 × 10^6^ cell/mL, HTR-8 = 1.7 × 10^6^ cell/mL, and HuH-7 = 0.85 × 10^6^ cell/mL.

Viral Kinetics ZIKV-IEC-Paraíba. The titer of the viral stocks was 1.5 × 10^6^ PFU/mL. We used MOI = 0.5 for ZIKV-IEC, with the following averages of cell concentration: BeWo = 2.3 × 10^6^ cell/mL, BeWo + forsk = 1.7 × 10^6^ cell/mL, HTR-8 = 1.6 × 10^6^ cell/mL, and HuH-7 = 0.32 × 10^6^ cell/mL.

The data obtained in our study should be analyzed independently for each ZIKV strain as we used different MOIs due to the lack of ZIKV-IEC titers to reach MOI = 1. Regarding BeWo lineage, due to the addition of forskolin, we performed the cultivation of treated and untreated cells two days before, and such a compound was added one day before inoculation. Eight assays of viral kinetics were fulfilled: four for each ZIKV strain, in triplicates, with seven hpi. Considering triplicates, there were a total of 208 samples, including negative controls for each hpi from intra and extracellular media (Apendix C).

For standardization, we quantified the C_T_ of viral RNA for all hpi of triplicate one. Based on these results, we also quantified triplicates two and three but only for three time-points: 24 h, 48 h, and 72 h. Therefore, in the end, the quantification of 96 samples was obtained by the C_T_ result converted into PFU/mL. Since the results obtained after the analysis of triplicates two and three validated those of triplicate one, we proceeded the study (Figure 1, step 1 and 2). The complete conversion table is found in (Appendix A).

In general, all BeWo, HTR-8 and HuH-7 controls presented similar features in all hpi. At 2 and 24 hpi, the monolayer was intact and the cells had visible boundaries and the absence of deformities, cell division, observable nucleus, and nucleolus. At 48 hpi, the monolayer was more closed, with the appearance of cells in the supernatant (refringent cells), and also the presence of characteristics found in the previous hpi. Furthermore, at and hpi between 72 and 144, there was saturation of the space occupied by the monolayer, augmentation of the cells in the supernatant, and the same characteristics of 48 hpi (Apendix I).

On the other hand, BeWo/HTR-8/HuH-7-infected cells displayed cytopathic effects (CPEs) that increased during the time points analyzed. Monolayer detachment, focal degeneration with rounded and refractory cells, partial and total destruction of the monolayer inoculated, as well as the formation of cellular debris, morphological alterations, swelling, and clusters of cells could be observed (Figure 2a,b). Thus, we considered the three cell lineages susceptible to the ZIKV-MR766lp and ZIKV-IEC viruses.

### 3.1. Inoculation of ZIKV-MR766lp (a)

Regarding the kinetics curve after infection with ZIKV-MR766lp (Figure 3 and Apendix I), BeWo and BeWo + fork had the following similarities: peak of intracellular viral production between hpi 24 and 48, in addition to a decay soon after this period. Considering the presence of the virus in the extracellular medium, there was growth in the first three time points. After these periods, a constant stability remained until the end. 

BeWo showed a fast proliferation, although the onset of the monolayer remained with low density. After 3–4 days, 100% confluence was observed, with the detachment of older cells (refringent cells). Concurrently with the CPE, the cell growth was maintained at the beginning of the kinetics. HTR-8 showed similar dynamics to the other cell types. At 24 and 48 hpi, there was a peak of intracellular viral production, followed by a rapid decline. In contrast, the extracellular medium remained constant throughout the kinetics. Furthermore, after 72 hpi, there were plenty of cells still in the monolayer. HuH-7 presented a peak of intracellular viral production after 24 h hpi and continued to decline until 144 hpi. The extracellular medium showed a lot of cellular debris after the peak of infection (Figure 2a and Appendix A).

### 3.2. Inoculation of ZIKV-IEC-Paraíba (b)

After ZIKV-IEC-Paraíba infection, BeWo and BeWo + fork showed different growth curves (Figure 3 and Appendix J). The amount of PFU/mL of BeWo cells remained higher intracellularly until 120 hpi, and the peak production occurred at 48 hpi. Regarding the extracellular medium, the PFU/mL was maintained lower intracellularly until 144 hpi, when it became higher. The CPE was observed at 72 hpi and augmented at 144 hpi, but part of the monolayer retained the characteristics of non-infected cells (Appendix A). BeWo + fork showed intense viral production between 24 and 72 hpi, with a drop at 96 hpi and an increase between 120 and 144 hpi. During all the kinetics, the amount of virus in the extracellular medium remained constant, with few fluctuations over time. The CPE was observed at 48 hpi and progressively increased with hpi. Furthermore, there was monolayer detachment and cellular debris (Figure 3 (bottom) and Appendix A).

HTR-8 maintained a high PFU/mL in the intracellular environment until 96 hpi. In contrast, at 120 and 144 hpi, the number of virions augmented in the extracellular medium and decreased in the intracellular medium, probably indicating cell bursts. The CPE started was observed at 96 hpi, but debris formation only occurred at the next hpi. Most cells remained adhered to monolayers and showed signs of infection (Figure 3 (bottom) and Appendix A).

HuH-7 viral production in intracellular and extracellular medium was high between 24 and 72 hpi. A fast decline of growth occurred in the intracellular environment after this period, whereas the extracellular environment remained constant until 144 hpi. The CPE became apparent at 48 hpi and increased up to 144 hpi, when all cells formed cellular debris and there was no longer a monolayer (Figure 2b and Appendix A).

We considered seven time points to describe the viral infection and, its capacity to generate new infectious progenies and to observe the CPE. After six days, at 144 hpi for ZIKV-MR766lp, the majority of the available cells were infected, which reflected in the considerable decay of alive cells. Regarding ZIKV-IEC, cell death by infection was only high for HuH-7. While in BeWo, BeWo + fork, and HTR-8 alive cells persisted—possible abortive infection? Moreover, BeWo + fork and HTR-8 cells sustained the infection and a possible new round of viral replication, even after the infectious peak. Another fact noticed in the kinetics, for all assays with the exception of HuH-7, is that the cell monolayer remained the same in the last three hpi, when compared to its non-inoculated controls. The most prominent CPEs in infected cells were morphological changes and death. Dead cells were also present in the non-inoculated control, due to the fact that the control trophoblastic cells reached maximum confluence and lost space during monolayer expansion. These results may be an indication that during the replicative cycle of the virus, there was no inhibition of cell multiplication. Moreover, in the kinetics end, the supernatant had dead cells by both the virus and cells, possibly alive, due to the continuity of monolayer.

## 4. Discussion

Considering the results obtained by us, all cell lineages are permissive and susceptible to both lineages of ZIKV with common characteristics CPEs observed in *Flavivirus*.

Regarding virus transmission by the placental barrier, the first 12 gestational weeks are the most critical, because the placenta is still developing and its protection depends on maternal antibodies [11,12,13,14,17,18,19,33,34,35]. Among the proposed routes, virus transmission can occur via trophoblasts [33,34,35], which are pluripotent cells that arise in the blastula stage and the beginning of the first cell differentiation and will originate from the embryo and embryonic annexes [36]. Studies were published about the development of ZIKV infection in embryonic stem cells, from the first gestational semester, and in human placental explant [33,34,35,37,38,39]. These papers comparatively showed how the two strains infect and cross the placental protective barrier and showed that both lineages are permissive and susceptible. However, they did not show why the Asian lineage can cross the placental barrier and causes CZS infection in the embryo/fetus/newborn while African lineage did not.

The percentage of maternal fetal transmission is estimated by 20–30% of pregnant women [9]. The hypothesis states that the maternal fetal route of transmission, excluding the primary route of transmission by vector or sex, occurs through the placental tissue. Villous trophoblast cells (VT), present in the beginning of embryogenesis, differentiate into cytotrophoblasts (CTBs) and then into syncytiotrophoblasts (STBs), which correspond, respectively, to BeWo and BeWo + fork. Forskolin induces BeWo to form syncytial cells. Based on the VT, there is a subset of extravillous cells (EVTs) that matches with the HTR-8 lineage. Such cells arise between the sixth and twelfth week, in the first trimester of pregnancy, and constitute an efficient barrier against microorganisms like viruses. Thus, since the onset of gestation, the interaction between the maternal uterine and embryonic placental tissues promotes a barrier against the entry of microorganisms [37,39]. The first trimester is also considered the period of greatest risk for the embryo/fetus to contract CZS [9,12,33,34,35,37,39,40]. Another factor of gestational vulnerability in the first trimester is that maternal IgG crosses the placental barrier only from the second trimester of gestation [41,42].

The comparison of African and Asian lineages in vitro (in different cell types), in vivo, in non-human primates, and in the most common species of *Aedes* spp. is presented and discussed in the scientific literature [7,8]. The majority of such experiments showed that the African strain is more infectious than the Asian strain [38,43,44,45,46,47,48,49]. However, only the Asian strain has the potential to cause CZS and GBS [8,11,12,45]. 

Considering the hypotheses that emerged for ZIKV to have become an infectious agent capable of infecting neural progenitor cells (NPCs), one is due to point mutations in structural genes D67N [43] and S139N [44], and non-structural T2634V [39]. When considering ZIKV-MR766 (African), ZIKV-Asian (2010) and ZIKV-Asian (2013–2015) genomes, through existing in vivo and in vitro experiments such mutations are at least considered hot-spots [43,44,45,46]. Such mutations are related to increased infectivity, cellular tropism for NPC and NS cells, and less virulence, but with greater persistence in tissues [44]. Considering that in our study, the BeWo and HTR-8 cells showed similar dynamics, it is possible that these same genes also favor viral tropism in placental cells and infect the embryo/fetus. However, further study is needed.

Another hypothesis is that the target cells can be modified by ZIKV and induce apoptosis, necrosis, and paraptosis [9]. During viral cycle replication in trophoblast cells, ER suffers stress that can trigger mechanisms of apoptosis [50]. Considering the results of the Asian-Brazilian lineage, one supposes that the lineage cells BeWo and HTR-8 remain alive for longer, maybe because the apoptosis pathway is not activated or is blocked for some reason; this has not been investigated yet.

Our study is based on the models by Delbruck and Ellis, and Burleson and has as its mainly aims to bring about an improvement by applying mathematical calculations to quantify virus production originating from bench experiments. Carrera et al. [19] compared ZIKV with WNV (West Nile virus), quantified each time point by PFU/mL, and built a curve graph with this result. In our work, we improved this method and established a relationship between C_T_ values obtained by qRT-PCR and the plaque assay (PFU/mL). In this way, we have a standard curve that can be used to convert the values in C_T_ for each time point into PFU/mL without performing plaque assays for all time points and the replicas. We can in the future combine this approach with statistical models and other methods as, for instance, High-content Screening/Analysis (HCS).

A practical application combining our method with HCS would be antiviral drug screening, since there are still no antiviral drugs against ZIKV approved for human usage. Moreover, there are potential antiviral drugs that have already been tested with cells such as HuH-7 and Vero. Some of them have the potential to be used during pregnancy [51,52,53,54,55,56].

## 5. Conclusions

In summary, our study shows that first-trimester placenta cells are permissive and susceptible to African and Asian-Brazilian ZIKV infections in vitro. While ZIKV-MR766lp effectively infects placental cells, leading to fast death, which can be an indication that infection by such a strain is not persistent and is less transmissible to fetus, ZIKV-IEC-Paraíba presented a sustained and longer replicative cycle, without inducing cell death. 

Although it is emphasized here that this type of test does not represent the conditions of the uterine environment, our method allows a better observation of the dynamics of ZIKV infection and, as such, should be encouraged for further in-depth studies.

## Figures and Tables

**Figure 1 viruses-15-01446-f001:**
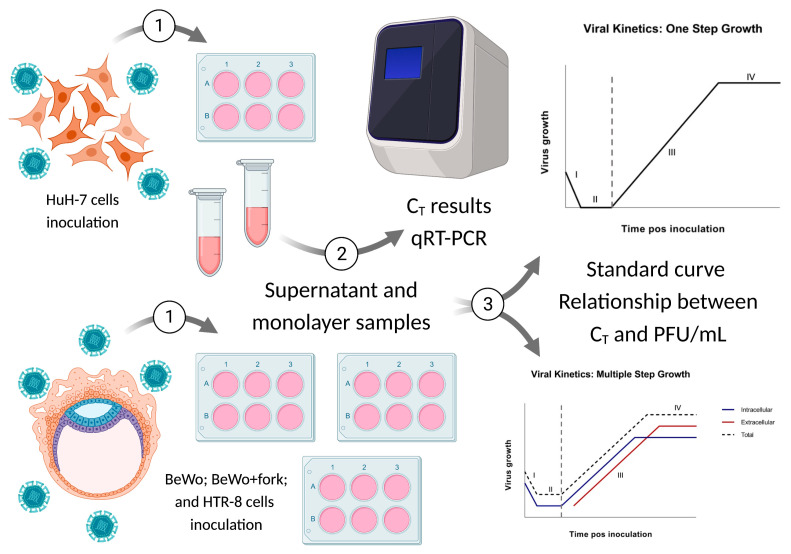
Experiment with viral kinetics. Steps 1 and 2 represent cell cultures in plates of six-wells plates where they were inoculated with ZIKV and collated after each hour post infection (hpi), and the monolayer and supernatant underwent qRT-PCR analysis. In step 3, the viral growth curve was elaborated by Ellis and Delbruck [20], whereas Burleson [23] presented the graphical description of the phases of the replicative cycle related to enveloped viruses, considering the time versus unit of infectious particles formed. Based on the models by Delbruck and Ellis, and Burleson, the curve can be divided into two parts. The first is characterized by the (I) adsorption, and (II) penetration and disassembly of viral particles: onset period of transcription, translation, and replication of the viral genome. In this stage, few viruses are detectable, and the most accurate verification of their presence is by means of qRT-PCR or immunofluorescence. The second part involves the assembly, (III) maturation and (IV) release of the viral progeny, in addition to its detection by different molecular and cellular methods. Both phases occur concurrently.

**Figure 2 viruses-15-01446-f002:**
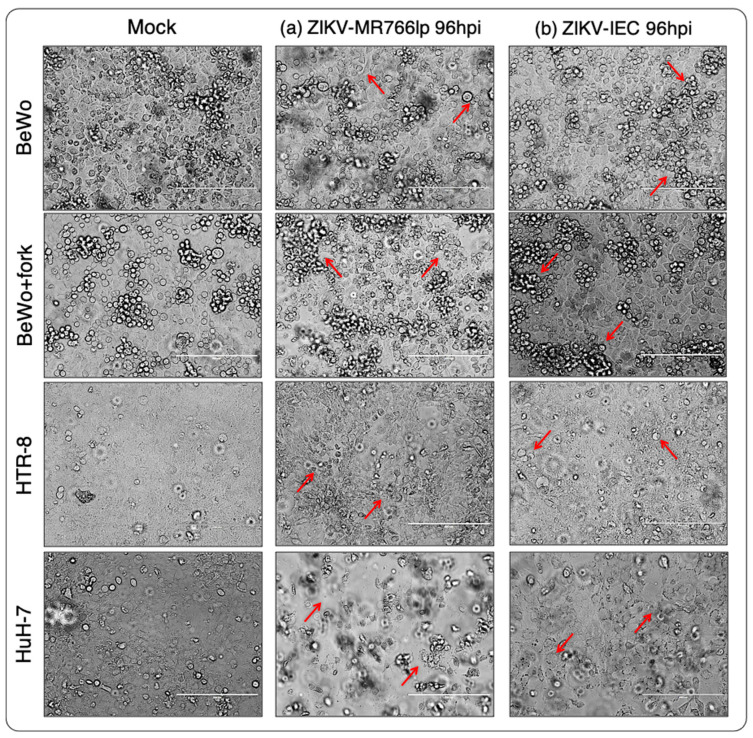
Image obtained by inverted-light optical microscope with magnification at 200 μm (20×). The data obtained in our study should be analyzed independently for each ZIKV strain as we used different MOIs due to the lack of ZIKV-IEC titers to reach MOI = 1. The monolayer of infected cells (red arrow), from the viral kinetics experiment, of (**a**) ZIKV-MR766 low passage (African) and (**b**) ZIKV-IEC-Paraíba (Asian-Brazilian) strains hpi 96 h post infection. Comparative observation between mock and infected cells showed the characteristic cytopathic effects (CPEs) of ZIKV. Furthermore, such features seen, under the optical microscope, in the BeWo, BeWo + fork, and HTR-8 were the same, as noticed in the HuH-7 control lineage: monolayer detachment; focal degeneration with rounded and refractory cells; partial and total destruction of monolayer inoculated; generation of cellular debris; morphological alterations, edema, and crowding of cells. The non-infected BeWo lineage, as observed in its the daily maintenance of it, began to detach from the monolayer after three days. Such cells had rapid cell division and expansion. When they reached 100% of space occupation, the oldest spontaneously separated from the monolayer.

**Figure 3 viruses-15-01446-f003:**
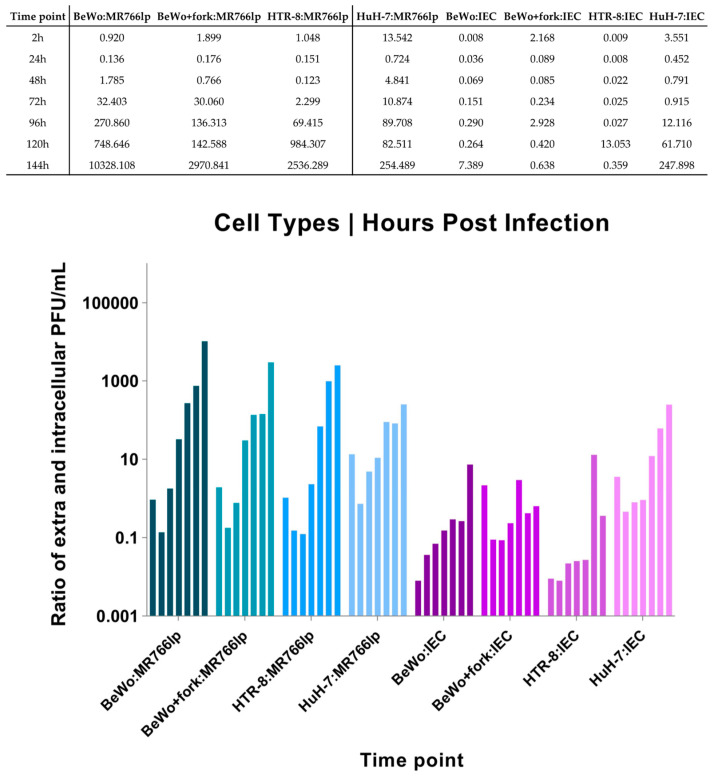
Rate graphic of PFU/mL relation between extra- and intracellular viral kinetics of BeWo, BeWo treated with forskolin, HTR-8, HuH-7 infected by ZIKV-MR766 low passage, and ZIKV-IEC-Paraíba (Appendix H). The abscissa shows the infected lineages during the period of post infection (hpi) from 2 to 144, while the ordinate indicates a relation between extra- and intracellular Log_10_PFU/mL. The data obtained in our study should be analyzed independently for each ZIKV strain as we used different MOIs due to the lack of ZIKV-IEC titers to reach MOI = 1.

## Data Availability

Not applicable.

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
