# Peer review of "Proposal of Model for Evaluation of Viral Kinetics of African/Asian/Brazilian—Zika virus Strains (Step Growth Curve) in Trophoblastic Cell Lines"

_viruses, 2023, doi:10.3390/v15071446_

Round 1
Reviewer 1 Report
Duarte-Barbosa et al. present a study that aims to investigate the dynamics of trophoblast cells during the first trimester of pregnancy. The research is highly relevant to the field and addresses a significant knowledge gap.
1: However, it should be noted that similar models have been previously evaluated in Carrera J., 2021 Virology Journal; Xu Xu. H., 2022 Vaccines; and Muthuraj GP., 2021 Cell Death Discovery. I recommend that the authors cite these papers to acknowledge the existing literature in the field.
2: Furthermore, the authors should provide a more detailed discussion on the novelty of their work. Clearly articulating the unique aspects or contributions of their study will enhance the manuscript.
3: In the manuscript, the authors mentioned that "both cell types continued alive during the process of viral replication." However, the viability of these cells upon ZIKV infection has not been demonstrated, which is an important aspect of establishing a cell model. I strongly recommend conducting a viability assay to determine the percentage of live or dead cells following ZIKV infection.
4: Regarding Figure 2, where microscopic images are included to demonstrate the cytopathic effect (CPE) induced by ZIKV, it is unclear how to differentiate between mock and ZIKV-infected cells. To improve clarity, I suggest the authors use arrows or other markers to highlight the CPE. Additionally, acquiring good-quality images will greatly assist in understanding the findings.
5: Lastly, I am curious whether the authors have screened any known drugs to inhibit ZIKV in these cells. Exploring the potential effects of existing antiviral agents would be a valuable addition to the study.
Minor editing of English language required
Reviewer 2 Report
The manuscript titled "Proposal Model for Evaluation of African/Asian/Brazilian- 2 ZIKV Strains Viral Kinetics (Step Growth Curve) in Tropho- 3 blastic Cell Lines'. has been written very well where authors want to show the three placental cells lines that are permissive and susceptible to ZIKV.
Minor issue.
Title of figure 3 is misspelled. hour pos infection. It should be hour post infection.
minor editing required.
Round 2
Reviewer 1 Report
The revised manuscript looks fine to me.